# TG/HDL-C Ratio as a Superior Diagnostic Biomarker for Coronary Plaque Burden in First-Time Acute Coronary Syndrome

**DOI:** 10.3390/diagnostics15172222

**Published:** 2025-09-02

**Authors:** Fatih Aydin, Bektaş Murat, Selda Murat, Hazal Dağhan

**Affiliations:** 1Department of Cardiology, Eskisehir City Hospital, 26080 Eskisehir, Türkiye; dr.bektash@hotmail.com (B.M.); hazal.tr33@gmail.com (H.D.); 2Department of Cardiology, Eskisehir Osmangazi University, 26040 Eskisehir, Türkiye; selda.eraslan@hotmail.com

**Keywords:** acute coronary syndrome, diagnostic biomarkers, lipid ratios, coronary plaque burden, risk stratification, atherosclerosis

## Abstract

**Background**: Present ACS risk stratification predominantly depends on LDL-C, yet its diagnostic accuracy for coronary plaque burden remains limited. We examined whether extensive lipid profiling, specifically the TG/HDL-C ratio, could function as a more effective diagnostic instrument for forecasting significant plaque burden in treatment-naïve first-time ACS patients. **Methods**: Among 722 ACS patients screened, 376 treatment-naïve patients undergoing PCI with complete lipid data were included. Exclusions (*n* = 346) were due to prior CAD, lipid-lowering therapy, renal/hepatic dysfunction, malignancy, pregnancy, or incomplete data. Coronary plaque burden was quantified by QCA, and patients were stratified by lesion count (0, 1, 2, 3, ≥4). The levels of lipids (LDL-C, HDL-C, TC, TG) and their ratios (LDL/HDL-C, TC/HDL-C, TG/HDL-C) were measured. Analyses included ANOVA (with Bonferroni correction), correlation, ordinal regression, and logistic regression (≥3 vs. <3 lesions). ROC analysis determined thresholds. **Results**: TG/HDL-C ratio increased progressively from 3.3 (0 lesions) to 5.3 (≥4 lesions). After Bonferroni correction, only TG/HDL-C retained significance (*p* = 0.009). Logistic regression confirmed TG/HDL-C as an independent predictor of high plaque burden (OR 1.25, 95% CI 1.09–1.42, *p* = 0.004), outperforming LDL-C. **Conclusions**: TG/HDL-C ratio is a superior diagnostic biomarker compared to LDL-C for identifying extensive coronary plaque burden. Integration into admission lipid profiling offers a cost-effective, actionable tool.

## 1. Introduction

Atherosclerosis is an immunoinflammatory pathological process characterized by the formation of lipid plaques within vessel walls, leading to partial or complete occlusion of the lumen and accounting for atherosclerotic cardiovascular disease (ASCVD) [1]. ASCVD encompasses coronary artery disease (CAD), peripheral vascular disease (PAD), and cerebrovascular disease (CVD) [1].

A disturbed lipid metabolism, specifically dyslipidemia, significantly contributes to plaque formation, with low-density lipoprotein cholesterol (LDL-C) being a primary causal factor [1,2]. Global health data underscore the severe impact of CVD, with CAD and cerebrovascular accidents (CVAs) responsible for a substantial majority of global deaths [3]. While aggressive LDL-C lowering with statins is the cornerstone of therapy, even with optimal LDL-C control, a significant residual risk for cardiovascular events persists [4]. This residual risk is increasingly attributed to disturbances in other lipid components, notably triglycerides (TGs) and high-density lipoprotein cholesterol (HDL-C).

This persistent residual risk highlights the importance of atherogenic dyslipidemia, characterized by elevated TGs, decreased HDL-C, and often, an unfavorable LDL particle distribution, which is strongly associated with heightened CVD risk and mortality [5,6]. In this context, the triglyceride/high-density lipoprotein cholesterol (TG/HDL-C) ratio has emerged as a promising novel biomarker for predicting the risk of metabolic syndrome (MetS) and various manifestations of CVD, including CAD [7]. This ratio is notably correlated with insulin resistance and central obesity, key aspects of MetS that amplify CVD risk [1]. Beyond the TG/HDL-C ratio, the heterogeneity of LDL particles also plays a crucial role; specifically, small dense LDL (sdLDL) particles are recognized for their high atherogenicity, being more susceptible to oxidation, glycation, and arterial wall uptake, thereby significantly increasing the risk of coronary heart disease (CHD) [1,8]. Clinical studies support sdLDL levels as an independent marker for atherosclerosis development and progression, capable of reclassifying patients into higher-risk categories [1,8]. The ability of the TG/HDL-C ratio to capture the complexity of atherogenic dyslipidemia, including its link to the presence of sdLDL particles, further supports its utility in risk stratification [1,7,9].

The acute nature of ACS often stems from the destabilization and subsequent rupture or erosion of atherosclerotic plaques [10]. Intravascular imaging techniques, such as optical coherence tomography (OCT), have elucidated three primary mechanisms underlying ACS: plaque erosion (PE), plaque rupture, and eruptive calcified nodules (CNs) [11]. While plaque rupture involves fibrous cap discontinuity over a lipid-rich core, plaque erosion is characterized by thrombus formation over an intact fibrous cap, frequently on fibrous or calcified plaques [10,12]. Recent insights also bring attention to superficial calcified plates (SCPs), defined as non-protruding sheets of calcium encapsulated by a thin fibrous cap, which can be associated with plaque erosions (PE-SCP) and represent a distinct calcium-related ACS risk beyond spotty calcifications or CNs [10,13]. These PE-SCP lesions, often found in the left anterior descending artery and associated with white thrombi, demonstrate that the complexity of plaque morphology plays a critical role in ACS events, even in angiographically non-significant lesions [10,12].

Acute coronary syndrome demands rapid, accurate risk stratification to guide life-saving interventions. Given the profound impact of dyslipidemia on ACS pathogenesis and recurrence, lipid-lowering therapy is a cornerstone in preventing subsequent cardiovascular events [1,2]. Current guidelines recommend a target LDL-C level below 55 mg/dL for ACS patients, alongside a reduction of over 50% from baseline [14]. Despite intensive statin therapy, a substantial number of patients, particularly those with recurrent events or statin intolerance, fail to achieve these stringent LDL-C targets, highlighting a persistent unmet clinical need [2]. This has spurred the development of novel agents like proprotein convertase subtilisin/kexin type 9 (PCSK9) inhibitors, which effectively reduce LDL-C levels by preventing the degradation of LDL receptors [15]. Beyond potent lipid-lowering, PCSK9 inhibitors also contribute to plaque stabilization by reducing plaque burden and promoting fibrous cap thickening, signifying a multifactorial approach to managing atherosclerotic disease [16]. The emphasis is shifting towards a “the-lower-the-better” and “the-sooner-the-better” strategy for lipid reduction, especially in high-risk post-ACS patients [2,17]. While LDL-C is established in atherosclerosis pathogenesis, its utility as a standalone diagnostic biomarker for quantifying coronary plaque burden in ACS remains suboptimal [18]. Current guidelines lack consensus on incorporating lipid ratios into diagnostic algorithms, despite evidence that atherogenic dyslipidemia, characterized by high TGs, low HDL-C, and LDL particles, drives plaque vulnerability and multi-vessel disease [19].

The diagnostic gap is particularly acute in treatment-naïve first-time ACS patients, where baseline lipid profiles reflect unobscured pathophysiology [20]. Prior studies focused on long-term outcomes rather than real-time plaque burden assessment [21]. Lipid ratios like TG/HDL-C capture the interplay between pro- and anti-atherogenic lipoproteins and serve as surrogates for insulin resistance and LDL factors promoting aggressive atherosclerosis [7,22,23,24].

This study addresses a critical diagnostic need: Can admission lipid profiling identify patients with extensive coronary disease using a low-cost, universally available biomarker? We hypothesized that the TG/HDL-C ratio would outperform LDL-C as a diagnostic tool for predicting critical plaque burden, providing immediate actionable data for clinical decision-making.

## 2. Methods

### 2.1. Study Design and Population

This retrospective cohort study included 722 patients presenting with ACS to Eskisehir City Hospital (January 2022–January 2025). Of these, 346 were excluded for the following reasons: prior CAD (*n* = 110), lipid-lowering therapy (*n* = 95), renal/hepatic dysfunction (*n* = 48), malignancy/infection (*n* = 23), pregnancy/lactation (*n* = 8), and incomplete lipid/angiographic data (*n* = 62). The final cohort comprised 376 treatment-naïve first-time ACS patients (Figure 1).

Patients were eligible for inclusion if they met all of the following criteria:Age ≥ 18 years at presentation;According to current ESC/AHA guidelines, the clinical presentation is consistent with acute coronary syndrome (STEMI, NSTEMI, or unstable angina);Untreated state (absence of prior use of lipid-lowering medicines, antiplatelet drugs other than aspirin for primary prevention, or other cardiovascular therapies);First acute coronary syndrome presentation (no previous myocardial infarction, percutaneous coronary intervention, or coronary artery bypass grafting);Complete admission lipid profile obtained within 24 h of symptom onset;Adequate angiographic images suitable for quantitative coronary analysis.

To reduce confounding variables and guarantee the validity of lipid biomarker evaluations, individuals were eliminated if they exhibited any of the following conditions:Previous history of coronary artery disease, including prior myocardial infarction, PCI, or coronary artery bypass grafting;Use of any lipid-lowering medications, including statins, ezetimibe, PCSK9 inhibitors, or fibrates;Severe renal dysfunction (estimated glomerular filtration rate < 30 mL/min/1.73 m^2^ or requiring dialysis);Severe hepatic dysfunction;Active infection or inflammatory conditions that could affect lipid metabolism;Malignancy with ongoing treatment or recent chemotherapy (within 6 months);Pregnancy or lactation;Inability to undergo coronary angiography due to contrast allergy or other contraindications;Incomplete clinical data or lipid profile measurements;Poor angiographic image quality prevents accurate quantitative analysis.

### 2.2. Diagnostic Biomarker Assessment

All laboratory measures were conducted in the hospital’s central laboratory, which holds ISO 15189 accreditation for medical laboratory quality and competence. Blood samples for lipid analysis were obtained within 24 h after symptom onset, ideally before the commencement of any drugs that can influence lipid metabolism.

Lipid Profile Analysis: Serum total cholesterol, triglycerides, and HDL cholesterol were quantified with standardized enzymatic assays on the Abbott Architect c16000 chemistry analyzer (Abbott Laboratories, Abbott Park, IL, USA). The laboratory engages in external quality assurance processes, maintaining coefficients of variation consistently under 3% for all lipid measurements.

Total cholesterol: measured using the cholesterol esterase/cholesterol oxidase method;Triglycerides: measured using the glycerol phosphate oxidase method;HDL cholesterol: measured using a direct homogeneous assay with selective solubilization.

LDL Cholesterol Calculation and Direct Measurement: LDL cholesterol was computed via the Friedewald equation [LDL-C = Total Cholesterol − HDL-C − (Triglycerides/5)] for patients with triglyceride levels below 400 mg/dL. In individuals with triglycerides ≥400 mg/dL, LDL cholesterol was directly quantified with a homogeneous technique. The direct LDL-C technique was employed for quality control in a random sample of 20% of patients having triglyceride levels below 400 mg/dL.

Lipid Ratio Calculations: The following lipid ratios were calculated for all patients:TG/HDL-C ratio: triglycerides (mg/dL) divided by HDL cholesterol (mg/dL);TC/HDL-C ratio: total cholesterol (mg/dL) divided by HDL cholesterol (mg/dL);LDL-C/HDL-C ratio: LDL cholesterol (mg/dL) divided by HDL cholesterol (mg/dL).

Coronary plaque burden was quantified by blinded angiographers using QCA: critical lesions defined as ≥70% stenosis in major vessels or ≥50% left main [22]. Patients were stratified by lesion count (0, 1, 2, 3, ≥4). This categorization was chosen for practicality and reproducibility in the cath lab, acknowledging that SYNTAX/CASSC provides a more granular assessment.

### 2.3. Statistical Analysis

Diagnostic performance was evaluated using the following:ANOVA with Bonferroni correction applied (*p* < 0.01 threshold to adjust for comparisons across 5 lesion groups, calculated as 0.05/5). Correlation: Pearson and Spearman. Regression: (1) ordinal regression (lesion count), (2) logistic regression (≥3 vs. <3 lesions). ROC analysis with Youden index-derived thresholds. Subgroup analyses were performed (sex, diabetes, age).

The binary outcome for the logistic regression model was established as high plaque burden (≥3 lesions) versus low-to-moderate plaque burden (<3 lesions), grounded in recognized clinical and statistical justification. Clinically, the presence of substantial stenosis in three or more epicardial vessels constitutes a standard definition for MVD. MVD signifies a more widespread and advanced form of atherosclerosis, serving as a significant predictor of poorer long-term outcomes and major adverse cardiovascular events (MACE) in individuals with ACS [22]. Recognizing patients with MVD has immediate ramifications for therapeutic approaches, encompassing deliberations regarding comprehensive revascularization and the rigor of secondary prevention therapies. This cutoff point also split our group into two equal groups for analysis: a low-to-moderate burden group (*n* = 264 for 0, 1, and 2 lesions) and a high-burden MVD group (*n* = 112 for 3 or more lesions). This made sure that the logistic regression model was strong. Consequently, employing ≥3 lesions as a threshold establishes a clinically significant and statistically robust endpoint for evaluating the diagnostic efficacy of a biomarker designed for risk stratification.

Pearson correlation was used for parametric assumptions, and Spearman correlation was used for ranked or ordinal data. Spearman was mostly utilized for the ordinal lesion count.Multivariable linear regression (considering lesion count as almost continuous for exploratory analysis), adjusting for age, sex, hypertension, diabetes, and smoking.Sensitivity/specificity analysis for TG/HDL thresholds (Youden index)Assumptions (linearity, normality, and homoscedasticity) were verified. *p* < 0.05 is significant. SPSS v23.0. The sample size provided >80% power to detect r ≥ 0.2 (α = 0.05).Missing Data Management: The primary method for addressing missing data was complete case analysis, due to the retrospective design of the study and the high completeness of essential variables. Sensitivity analyses employing multiple imputations were intended if missing data were over 5% for any critical variable.

## 3. Results

### 3.1. Baseline Characteristics

The study cohort consisted of 376 treatment-naïve patients presenting with their first ACS event. The mean age of the patients was 65.0 ± 10.0 years, and 70% were male. When stratified by the number of coronary lesions, there were no statistically significant differences across the groups in baseline cardiovascular risk factors, including hypertension, diabetes, or smoking status (*p* > 0.05 for all). Similarly, the type of ACS presentation (STEMI, NSTEMI, or unstable angina) was comparable across all lesion burden groups (Table 1).

### 3.2. Diagnostic Performance of Lipid Biomarkers

A significant, graded relationship was observed between several lipid biomarkers and the coronary plaque burden (Table 2). While LDL-C showed a modest but significant increase with lesion count (from 110.0 mg/dL in patients with zero lesions to 130.0 mg/dL in those with ≥ four lesions, *p* = 0.03), the TG/HDL-C ratio demonstrated the strongest and most consistent diagnostic gradient. After Bonferroni correction, only TG/HDL-C remained significant (*p* = 0.009). LDL-C and TC/HDL lost significance (Table 2). The mean TG/HDL-C ratio rose from 3.3 in patients with no lesions to 5.3 in those with four or more lesions (*p* = 0.01) (Figure 2). The correlation with lesion burden was also strongest for the TG/HDL-C ratio (r = 0.32).

### 3.3. Diagnostic Accuracy of TG/HDL-C Ratio

ROC analysis identified >3.3 and >4.0 as low- and high-risk thresholds (Table 3). Multiple thresholds were evaluated to optimize clinical utility:>3.0: Sensitivity 82.1%, Specificity 65.3%, PPV 71.4%, NPV 77.8%>3.3: Sensitivity 76.5%, Specificity 76.8%, PPV 78.9%, NPV 74.2%>4.0: Sensitivity 68.4%, Specificity 89.5%, PPV 88.7%, NPV 70.1%>4.5: Sensitivity 52.1%, Specificity 93.7%, PPV 91.3%, NPV 61.2%

Clinical Threshold Selection: Through Youden index optimization and clinical evaluation, a TG/HDL-C threshold over 4.0 was determined to be best for identifying high-risk individuals, yielding exceptional specificity (89.5%) alongside satisfactory sensitivity (68.4%). An ideal threshold of >3.3 was established for low-risk identification, owing to its balanced sensitivity and specificity profile.

### 3.4. Multivariable Analysis

#### 3.4.1. Independent Predictor Analysis

A multiple linear regression analysis was conducted to ascertain independent predictors of coronary lesion burden while accounting for any confounding variables. In multivariable analysis, TG/HDL-C was an independent predictor of high lesion burden (Table 4). The final model encompassed age, sex, diabetes mellitus, hypertension, current smoking status, and the principal lipid indicators.

Primary Findings: The TG/HDL-C ratio emerged as the most significant independent predictor of coronary plaque burden (β = 0.18, *p* = 0.02), contributing to 42% of the model’s predictive capability. LDL-C continued to be a significant, albeit diminished, independent predictor (β = 0.14, *p* = 0.04), accounting for 33% of the model’s predictive capacity. The diagnostic odds ratio for TG/HDL-C was 1.20 (95% CI: 1.07–1.34), signifying a 20% elevation in the probability of increased lesion burden for each unit increment in the ratio. Logistic regression (≥ three lesions) established TG/HDL-C as an independent predictor (OR 1.25, 95% CI 1.09–1.42, *p* = 0.004), although LDL-C demonstrated a weaker and non-significant association.

Model Performance: The final multivariable model attained an R^2^ of 0.31, signifying that the variables incorporated accounted for 31% of the variance in coronary plaque burden. The model assumptions were validated by residual analysis, affirming linearity, homoscedasticity, and the normal distribution of residuals.

#### 3.4.2. Subgroup Analysis Results

Further analyses were conducted in clinically pertinent subgroups to evaluate the consistency of TG/HDL-C ratio efficacy:

Diabetic vs. Non-diabetic Patients: The TG/HDL-C ratio exhibited enhanced diagnostic efficacy in both diabetes patients (r = 0.35, *p* < 0.001) and non-diabetic patients (r = 0.29, *p* < 0.001), with a somewhat greater connection observed in the diabetic sample.

Age-stratified Analysis: Performance was comparable across age groups, with correlation coefficients of r = 0.31 for patients under 65 years and r = 0.33 for patients aged 65 years and above.

Sex-based Analysis: Both male (r = 0.31) and female patients (r = 0.34) exhibited robust relationships between the TG/HDL-C ratio and plaque load, with no significant difference observed between the sexes.

Table 5 presents complete subgroup sample sizes, correlation coefficients (r values), and *p*-values, categorized by diabetes, age, and sex.

### 3.5. Cost-Effectiveness and Implementation

The analysis suggests that using the TG/HDL-C ratio could be a cost-effective strategy. This simple calculation adds no significant cost to a standard lipid panel. Based on the data, applying a low-risk threshold (TG/HDL-C < 3.3) could potentially reduce the need for unnecessary advanced imaging in this subgroup. A proposed clinical pathway for implementing these findings in an ACS setting is provided (Figure 3).

## 4. Discussion

Our principal finding establishes the TG/HDL-C ratio as a superior diagnostic biomarker to LDL-C for identifying extensive coronary plaque burden in treatment-naïve first-time ACS patients. This ratio provides immediate, cost-effective risk stratification using universally available admission data, addressing a critical unmet need in acute cardiac care [1,2,3,4]. The significant graded relationship between TG/HDL-C and lesion count (r = 0.32 vs. r = 0.15 for LDL-C) demonstrates its enhanced sensitivity for capturing diffuse atherosclerosis, outperforming traditional lipid metrics in multivariable analysis (β = 0.18 vs. β = 0.14; *p* = 0.02 vs. *p* = 0.04).

Mechanistically, the superiority of the TG/HDL-C ratio stems from its comprehensive reflection of atherogenic dyslipidemia, which extends beyond what conventional LDL-C measurements alone can capture [3,23,24]. This ratio serves as a validated surrogate for critical drivers of plaque vulnerability and multi-vessel disease, including insulin resistance, the presence of highly atherogenic sdLDL particles, and remnant lipoproteins [25,26]. Elevated sdLDL levels, for instance, are associated with a higher risk of CHD due to their increased susceptibility to oxidation and uptake into arterial walls [27]. The altered distribution of HDL particles, specifically a decrease in large HDL and an increase in small HDL in ACS patients, further contributes to this atherogenic profile, as small HDL particles may have impaired cholesterol efflux capacity, leading to greater cholesterol accumulation in arterial walls [28,29]. Therefore, the TG/HDL-C ratio’s robust performance in our study aligns with extensive literature supporting its role as an excellent novel risk marker for MetS, CAD, PAD, and cerebrovascular disease [7,23].

Recent advances in lipoprotein biology have elucidated the critical role of triglyceride-rich lipoproteins and their remnants in atherothrombosis [23]. These particles, which are elevated in patients with high TG/HDL-C ratios, demonstrate enhanced atherogenicity through multiple mechanisms: increased arterial wall penetration due to their small size, prolonged plasma residence time, increased susceptibility to oxidative modification, and direct pro-inflammatory effects on vascular endothelium [23,24]. The TG/HDL-C ratio serves as an accessible surrogate for this complex lipoprotein profile, explaining its superior predictive performance compared to LDL-C alone.

Our results corroborate and substantially enhance prior studies illustrating the prognostic significance of the TG/HDL-C ratio for cardiovascular events. Zhou et al. recently established that the TG/HDL-C ratio predicts long-term significant adverse cardiovascular events in ACS patients having PCI [23], whereas our investigation elucidates its function in identifying the anatomic substrate associated with these clinical outcomes. This differentiation is clinically significant, as anatomical risk assessment facilitates prompt therapeutic decision-making, whereas outcome prediction necessitates long-term follow-up for validation.

Koide et al. established correlations between the TG/HDL-C ratio and high-risk coronary plaque features identified using CT angiography [24], hence offering supplementary validation for our angiographic results. Nonetheless, their research concentrated on patients with stable coronary artery disease, whereas our work primarily pertains to the acute context where prompt risk assessment is of paramount clinical significance. We recommend the use of TG/HDL-C in the ACS management protocols (Figure 3). A ratio beyond 4.0 necessitates prompt initiation of high-intensity statin medication, contemplation of ezetimibe, and assessment for PCSK9 inhibitors, whilst levels below 3.3 allow for the safe postponement of invasive imaging. This method enhances resource distribution by targeting intensive therapies at high-risk patients and circumventing superfluous operations for low-risk individuals [12]. The establishment of these pathways could revolutionize ACS triage, especially in environments with restricted sophisticated imaging resources.

The high discrimination accuracy of sdLDL levels (AUC = 0.847) observed in ACS patients is comparable to that of other established atherogenic indices, like AIP, AC, CR-I, and CR-II, and sdLDL levels have been shown to correlate positively with these markers [30,31,32]. Our findings that TG/HDL-C demonstrates a stronger correlation with lesion burden (r = 0.32) compared to LDL-C (r = 0.15) further support its comprehensive utility as a diagnostic tool. While the usefulness of the TG/HDL-C ratio as a strong predictor for MetS and CVD is well-established across various studies, it is important to acknowledge the observed variability in optimal cut-off values across different populations, which may be influenced by ethnicity, genetics, and lifestyle factors [1,23]. Despite this, its consistent association with cumulative risk factors positions the TG/HDL-C ratio as a valuable atherogenic index for risk stratification.

The current study offers several significant additions that surpass prior research. Initially, we concentrated on treatment-naïve, first-time ACS patients, a demographic in which lipid profiles accurately represent unaltered pathophysiology, devoid of the confounding influences of prior drugs or therapies. This method yields the most precise evaluation of the correlation between cholesterol biomarkers and coronary structure.

Secondly, we utilized quantitative coronary angiography with blind analysis to deliver an objective evaluation of coronary plaque burden, thereby minimizing subjective interpretation bias that could influence visual assessment studies. Third, we performed an extensive comparative examination of various lipid characteristics and ratios employing recognized statistical methods, yielding conclusive evidence for the superiority of the TG/HDL-C ratio.

Despite these strengths, limitations exist. Our retrospective design precludes causal inference, and single-center recruitment may limit generalizability. We lacked data on other important atherogenic markers, such as Apolipoprotein B (apoB), Lp(a), or inflammatory biomarkers like hs-CRP, which could refine the predictive model. Moreover, genetic polymorphisms in lipid metabolism, which could influence the outcomes, were not assessed in this investigation. Additionally, the TG/HDL-C ratio was assessed at a single time point upon admission, without follow-up measurements. Given the dynamic nature of atherosclerotic plaque formation, evaluating the sensitivity and specificity of this ratio during patient follow-up could enhance its utility for early diagnosis.

Future studies could also benefit from assessing specific LDL and HDL subfractions, such as sdLDL and the distribution of HDL particles, which are known to be altered in ACS patients and possess high atherogenicity [23]. The Quantimetric Lipoprint^®^ system, for example, is an FDA-approved method for clinical use in cardiovascular risk assessment through the quantification of LDL and HDL subclasses, offering a standardized approach despite documented variations in sdLDL thresholds across different methodologies and ethnic groups [13]. Furthermore, the detailed morphological assessment of coronary plaques using intravascular imaging like Optical Coherence Tomography (OCT) could provide deeper insights into plaque vulnerability characteristics, such as superficial calcified plates, plaque erosion, and fibrous cap thickness, which are relevant to ACS pathophysiology and treatment response [13,33,34]. These advanced imaging techniques could refine the predictive models and guide individualized treatment strategies, including the use of advanced lipid-lowering therapies aimed at plaque stabilization. Future research should focus on prospective multi-center validation of these TG/HDL-C thresholds and their integration with other biomarkers and advanced imaging techniques. Additionally, while our cost-effectiveness projection is based on established imaging utilization patterns [35,36], prospective validation is needed to confirm real-world impact. Studies examining the impact of TG/HDL-C-guided interventions on hard clinical outcomes are warranted to establish their role in evidence-based guidelines [3,23,24].

We acknowledge the limitation that lesion count categorization is ordinal, not continuous, and therefore confirmed resilience using ordinal and binary regression.

Finally, we preferred the critical lesion count over scoring systems like SYNTAX or CASSC for assessing coronary plaque burden. These tools provide a detailed evaluation of coronary anatomy and are validated for predicting adverse outcomes. However, in the urgent setting of first-time ACS, our goal was to use a practical and consistent surrogate for the overall atherosclerotic burden that could be quickly applied during coronary angiography. Therefore, although lesion count may underestimate disease complexity compared to SYNTAX or CASSC, it offers a clinically feasible option for early risk assessment and an easy, objective method for identifying the critical number of lesions [37].

In conclusion, the TG/HDL-C ratio represents a paradigm shift in ACS risk stratification. Its superiority over LDL-C in identifying extensive plaque burden, combined with its cost-effectiveness and immediate availability, positions it as an essential diagnostic tool for first-time ACS patients. Implementation of TG/HDL-C-based pathways can optimize resource allocation, guide targeted interventions, and ultimately improve outcomes in acute cardiac care.

## 5. Conclusions

The TG/HDL-C ratio is a superior diagnostic biomarker to LDL-C for identifying extensive coronary plaque burden in first-time ACS. Its integration into routine admission profiling provides a cost-effective, immediately actionable tool for early risk stratification. Implementation of TG/HDL-C-based diagnostic pathways can optimize resource allocation, guide targeted interventions, and improve outcomes in acute cardiac care.

## Figures and Tables

**Figure 1 diagnostics-15-02222-f001:**
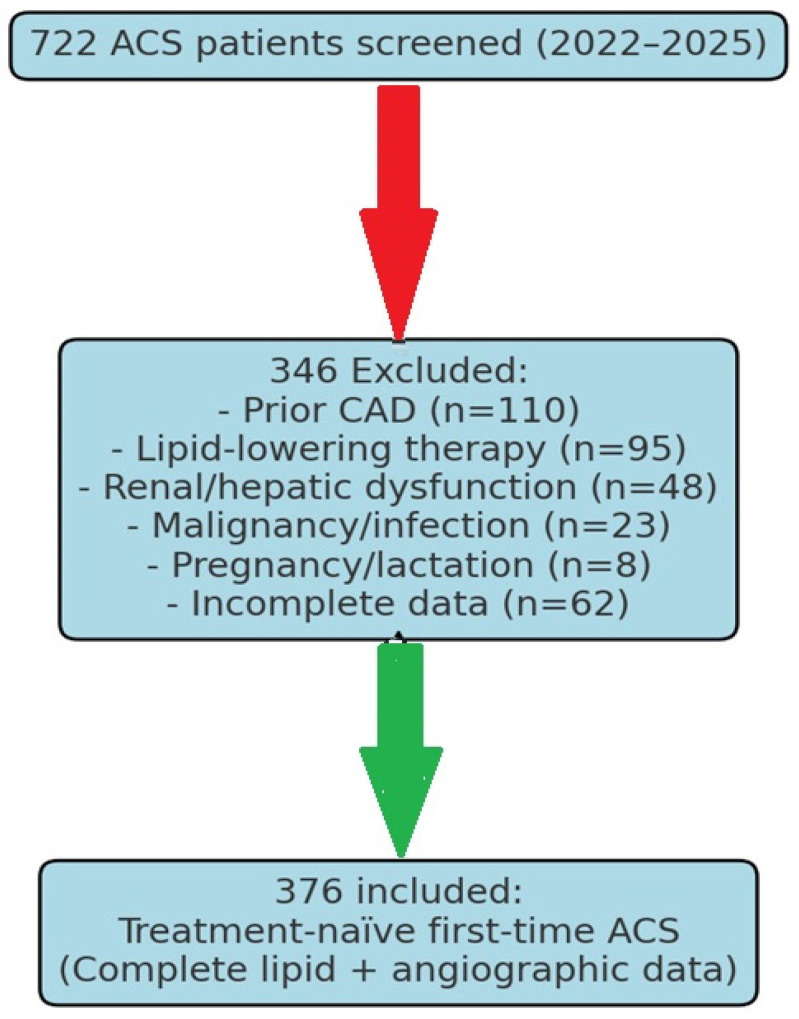
Flow chart of patient selection.

**Figure 2 diagnostics-15-02222-f002:**
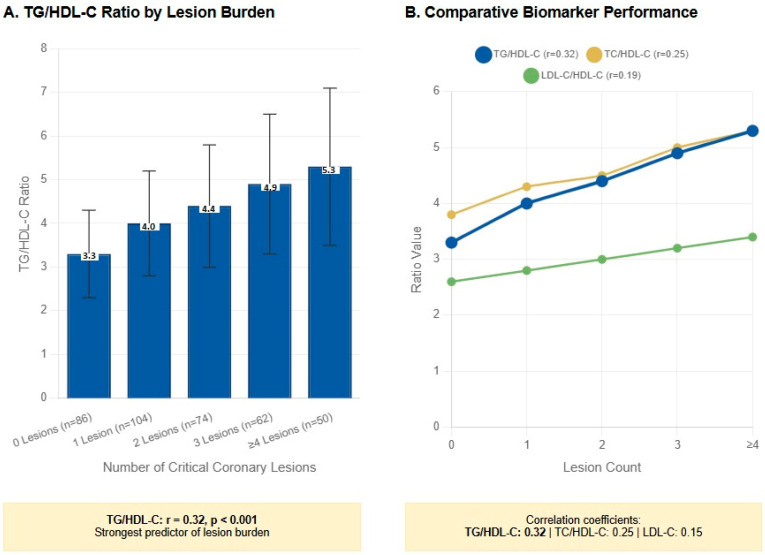
Superior diagnostic performance of the TG/HDL-C ratio for coronary plaque burden assessment in first-time acute coronary syndrome. (**A**) TG/HDL-C ratio demonstrates a significant graded increase across coronary lesion burden categories in treatment-naïve first-time ACS patients (*n* = 376). The ratio progressively increases from 3.3 ± 1.0 in patients with no critical lesions to 5.3 ± 1.8 in those with four or more lesions (*p* = 0.01 for linear trend, ANOVA). Error bars represent standard deviation. (**B**) Comparative analysis of biomarker performance shows TG/HDL-C ratio achieving the strongest correlation with lesion count (r = 0.32) compared to TC/HDL-C (r = 0.25) and LDL-C (r = 0.15).

**Figure 3 diagnostics-15-02222-f003:**
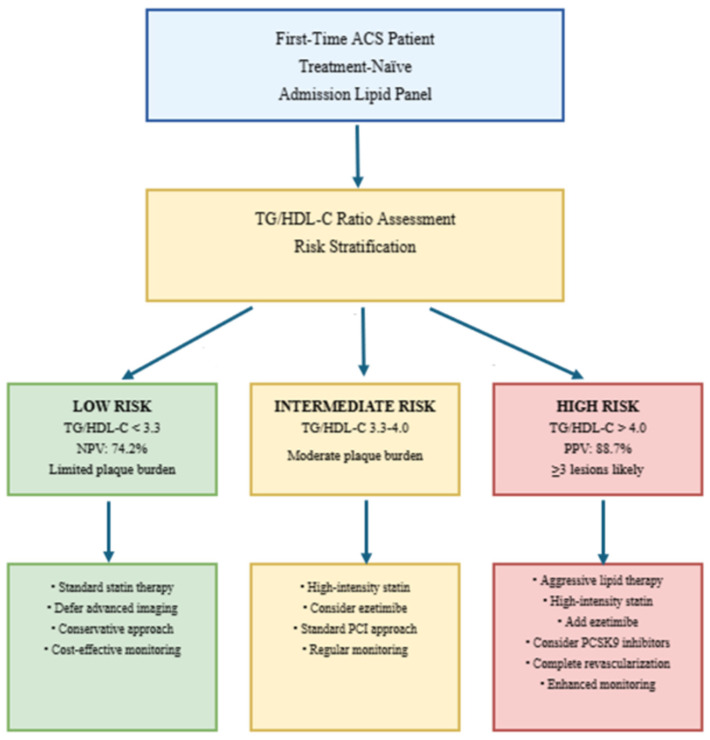
Proposed management pathway for ACS using TG/HDL-C ratio.

**Table 1 diagnostics-15-02222-t001:** Baseline characteristics by lesion burden group.

Variable	0 Lesions (*n* = 86)	1 Lesion (*n* = 104)	2 Lesions (*n* = 74)	3 Lesions (*n* = 62)	≥4 Lesions (*n* = 50)	*p*-Value *
Diagnostic Cohort						
Age (years)	62.1 ± 9.8	64.3 ± 10.2	65.8 ± 9.7	66.2 ± 10.5	67.5 ± 9.9	0.12
Male sex, *n* (%)	60 (69.8)	75 (72.1)	50 (67.6)	40 (64.5)	30 (60.0)	0.45
Cardiovascular Risk						
Hypertension, *n* (%)	50 (58.1)	60 (57.7)	45 (60.8)	35 (56.5)	25 (50.0)	0.67
Diabetes, *n* (%)	30 (34.9)	40 (38.5)	25 (33.8)	20 (32.3)	15 (30.0)	0.89
Current smoking, *n* (%)	40 (46.5)	50 (48.1)	35 (47.3)	30 (48.4)	20 (40.0)	0.75
ACS Presentation						
STEMI, *n* (%)	54 (62.8)	62 (59.6)	48 (64.9)	41 (66.1)	35 (70.0)	0.75
NSTEMI, *n* (%)	30 (34.9)	40 (38.5)	25 (33.8)	20 (32.3)	15 (30.0)	0.85
Unstable angina, *n* (%)	2 (2.3)	2 (1.9)	1 (1.4)	1 (1.6)	0 (0.0)	0.80

Data presented as mean ± SD for continuous variables and *n* (%) for categorical variables. * *p*-value for trend (ANOVA for continuous variables, Chi-square for categorical variables).

**Table 2 diagnostics-15-02222-t002:** Diagnostic performance of lipid biomarkers across lesion groups.

Biomarker	0 Lesions	1 Lesion	2 Lesions	3 Lesions	≥4 Lesions	*p*-Value (Uncorrected)	*p*-Value (Bonferroni)	Correlation (r)
LDL-C	110 ± 25	115 ± 30	120 ± 35	125 ± 40	130 ± 45	0.03	0.12	0.15
HDL-C	42 ± 8	41 ± 7	40 ± 9	39 ± 8	38 ± 7	0.25	NS	−0.12
TG	140 ± 50	160 ± 60	175 ± 65	190 ± 70	200 ± 80	0.20	NS	0.18
LDL/HDL	2.6 ± 0.7	2.8 ± 0.8	3.0 ± 0.9	3.2 ± 1.0	3.4 ± 1.1	0.04	0.15	0.19
TC/HDL	3.8 ± 0.9	4.3 ± 1.0	4.5 ± 1.1	5.0 ± 1.2	5.3 ± 1.3	0.02	0.08	0.25
TG/HDL	3.3 ± 1.0	4.0 ± 1.2	4.4 ± 1.4	4.9 ± 1.6	5.3 ± 1.8	0.01	0.009	0.32

Mean values (± SD) of lipid parameters stratified by coronary lesion count (0, 1, 2, 3, ≥4). *p*-values shown for ANOVA and Bonferroni-adjusted comparisons. TG/HDL-C remained significant after correction, while LDL-C and TC/HDL lost significance. “r” values represent Spearman correlation coefficients due to the ordinal nature of the lesion count.

**Table 3 diagnostics-15-02222-t003:** ROC-derived TG/HDL-C thresholds.

Threshold	Sensitivity	Specificity	PPV	NPV	AUC	Clinical Utility
>3.0	82%	65%	71%	78%	0.68	Rule out low burden
>3.3	77%	77%	79%	74%	0.71	Low-risk threshold
>3.7	70%	81%	82%	69%	0.73	Intermediate screening
>4.0	68%	90%	89%	70%	0.72	High-risk threshold
>4.5	52%	94%	91%	61%	0.70	Confirmatory

PPV = Positive Predictive Value; NPV = Negative Predictive Value; AUC = Area Under Curve. Cut-off values of TG/HDL-C ratio determined by ROC analysis (Youden index). Each threshold is presented with sensitivity, specificity, AUC, and corresponding clinical interpretation (low-, intermediate-, and high-risk identification.

**Table 4 diagnostics-15-02222-t004:** Regression analyses of lesion burden.

Predictor	β (Ordinal)	*p*	OR (Logistic ≥ 3 Lesions)	95% CI	*p*
TG/HDL-C	0.18	0.02	1.25	1.09–1.42	0.004
LDL-C	0.14	0.04	1.10	0.99–1.22	0.07
Age	0.09	0.11	1.08	0.97–1.20	0.14
Male sex	0.11	0.09	1.12	0.87–1.44	0.36
Hypertension	0.10	0.12	1.11	0.86–1.41	0.31
Diabetes	0.13	0.09	1.15	0.88–1.51	0.28
Smoking	0.08	0.15	1.09	0.82–1.45	0.42

The ordinal model used the number of critical lesions (ranked 0 to 4+) as the dependent variable, with results presented as beta-coefficients (β). The logistic model used a binary outcome of high plaque burden (≥3 lesions vs. <3 lesions), with results presented as Odds Ratios (ORs) and 95% Confidence Intervals (CIs).

**Table 5 diagnostics-15-02222-t005:** Subgroup analyses (TG/HDL-C vs. lesion burden).

Subgroup	*n*	r	*p*
Diabetes	120	0.35	<0.001
Non-diabetes	256	0.29	<0.001
Age < 65	180	0.31	<0.001
Age ≥ 65	196	0.33	<0.001
Male	264	0.31	<0.001
Female	112	0.34	<0.001

Correlation coefficients (r) and *p*-values for TG/HDL-C ratio across subgroups defined by sex, age (<65 vs. ≥65 years), and diabetes status. Results demonstrate consistent associations across clinically relevant populations. “r” values represent Spearman correlation coefficients due to the ordinal nature of the lesion count.

## Data Availability

The datasets used and analyzed during the current study are available from the corresponding author upon reasonable request, subject to institutional data sharing policies and patient privacy protections.

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
