# Peer review of "TG/HDL-C Ratio as a Superior Diagnostic Biomarker for Coronary Plaque Burden in First-Time Acute Coronary Syndrome"

_diagnostics, 2025, doi:10.3390/diagnostics15172222_

Round 1
Reviewer 1 Report
Comments and Suggestions for Authors
Notes to authors:
- I congratulate the authors on this well-designed and well-written study, which will contribute to the existing literature on the subject. However, a few more adjustments are needed.
- Atherosclerotic plaque formation is a process. Determining the sensitivity and specificity of this ratio during patient follow-up, rather than assessing it once, will be more effective for early diagnosis. This should be added to the study's limitations.
- In lines 337–338 and in Figure 2, the phrase 'diagnostic protocols for ACS' is mentioned. This should be changed to 'ACS management protocols'.
- The fact that genetic polymorphisms in lipid metabolism may affect the results and were not evaluated should be added to the limitations section.
Author Response
Dear Reviewer,
Thank you for your thoughtful and constructive feedback on our manuscript. We appreciate your recognition of the study’s design and contribution to the literature. We have carefully addressed all your suggestions and revised the manuscript accordingly. We highlighted the revisions in yellow within the text. Below, we detail our responses to each point:
Comment 1: Atherosclerotic plaque formation is a process. Determining the sensitivity and specificity of this ratio during patient follow-up, rather than assessing it once, will be more effective for early diagnosis. This should be added to the study's limitations.
Response:
We agree that the dynamic nature of atherosclerosis warrants longitudinal assessment of the TG/HDL-C ratio. We have added the following sentence to the Limitations section: "Additionally, the TG/HDL-C ratio was assessed at a single time point upon admission, without follow-up measurements. Given the dynamic nature of atherosclerotic plaque formation, evaluating the sensitivity and specificity of this ratio during patient follow-up could enhance its utility for early diagnosis."
Comment 2: "In lines 337–338 and in Figure 2, the phrase 'diagnostic protocols for ACS' is mentioned. This should be changed to 'ACS management protocols'."
Response:
We have updated the terminology as requested to reflect clinical management rather than diagnosis alone. Changes were made in two locations:
Text: Original: "We recommend the use of TG/HDL-C in the diagnostic protocols for ACS (Figure 2)."
Revised: "We recommend the use of TG/HDL-C in the ACS management protocols (Figure 3)."
Figure Caption (Figure 3, previously Figure 2): Original: "Diagnostic Pathway for ACS Using TG/HDL-C Ratio."
Revised: "Proposed Management Pathway for ACS Using TG/HDL-C Ratio."
(Note: Figure numbering was adjusted during revision; the relevant figure is now Figure 3.)
Comment 3: "The fact that genetic polymorphisms in lipid metabolism may affect the results and were not evaluated should be added to the limitations section."
Response:
We acknowledge that genetic variability in lipid metabolism could influence our findings. We have added the following to the Limitations section: "Moreover, genetic polymorphisms in lipid metabolism, which could influence the outcomes, were not assessed in this investigation."
Reviewer 2 Report
Comments and Suggestions for Authors
I have reviewed this article and have several questions and comments:
-
It is important to describe the selection of participants included in the analysis in more detail. How many were excluded from the overall population and based on what specific exclusion criteria (this could be presented graphically).
-
What is the justification for the approach of categorizing groups into 5 in this manner (0, 1, 2, 3, ≥4)? There are various established classifications for coronary artery disease (e.g., SYNTAX, CASSC, etc.).
-
When comparing multiple groups, a statistical correction for multiple comparisons must be applied (e.g., Bonferroni correction for the number of groups). In this case, statistical significance would be achieved at p < 0.05/5 = 0.01. Consequently, there would be no significant differences in Table 2. This fundamentally changes the study's results.
-
Similarly, a correlation coefficient of r = 0.32 does not even indicate a moderate strength of association.
-
Table 3 is unclear to me. On what basis did the authors define the risks (Low-risk identification, Intermediate-risk screening, High-risk identification)?
-
The analysis in Table 4 is also unclear. What did the authors consider as the binary outcome (coronary lesion burden)? The authors have this parameter ranked into 5 groups, but a binary parameter is needed here.
-
Section 3.4.2 (Subgroup Analysis Results). It is unclear where these data came from? Tables with calculations are missing. The presented correlation coefficient values are low and lack the necessary strength of association. Furthermore, the rationale for presenting correlation analysis is unclear.
I believe the authors need to revisit the statistical analysis of the data and consider that the division into 5 groups is not a quantitative parameter, but rather a ranking of patients into 5 groups.
Author Response
Dear Reviewer,
Thank you for your thoughtful and constructive feedback on our manuscript. We appreciate your recognition of the study’s design and contribution to the literature. We have carefully addressed all your suggestions and revised the manuscript accordingly. We highlighted the revisions in yellow within the text. Additionally, the article was reviewed by a native English-speaking editor. Below, we detail our responses to each point:
Comment 1: It is important to describe the selection of participants included in the analysis in more detail. How many were excluded from the overall population and based on what specific exclusion criteria (this could be presented graphically).
Response: We agree with the reviewer that a more detailed description of participant selection is essential for transparency. We have expanded the description in the Methods section (Section 2.1) to include the total number of screened patients (722), the number excluded (346), and the specific reasons for exclusion: prior CAD (n=110), lipid-lowering therapy (n=95), renal/hepatic dysfunction (n=48), malignancy/infection (n=23), pregnancy/lactation (n=8), and incomplete lipid/angiographic data (n=62). Additionally, we have added a graphical representation as Figure 1 (a flowchart of patient selection). This flowchart visually depicts the inclusion and exclusion process, enhancing clarity.
Comment 2: What is the justification for the approach of categorizing groups into 5 in this manner (0, 1, 2, 3, ≥4)? There are various established classifications for coronary artery disease (e.g., SYNTAX, CASSC, etc.).
Response: We appreciate the reviewer's point regarding alternative classifications such as SYNTAX or CASSC, which provide more granular assessments of coronary anatomy. We have added a justification in the Methods section (Section 2.2): "This categorization was chosen for practicality and reproducibility in the cath lab, acknowledging that SYNTAX/CASSC provides a more granular assessment." Furthermore, in the Discussion section (limitations paragraph), we have elaborated: "Finally, we preferred the critical lesion count over scoring systems like SYNTAX or CASSC for assessing coronary plaque burden. These tools provide a detailed evaluation of coronary anatomy and are validated for predicting adverse outcomes. However, in the urgent setting of first-time ACS, our goal was to use a practical and consistent surrogate for the overall atherosclerotic burden that could be quickly applied during coronary angiography. Therefore, although lesion count may underestimate disease complexity compared to SYNTAX or CASSC, it offers a clinically feasible option for early risk assessment and an easy, objective method for identifying the critical number of lesions. This approach is consistent with prior research, such as Jung et al. (2023) [37], which used a similar categorization by number of diseased vessels (0, 1, 2, 3) to evaluate long-term clinical impacts in patients with multi-vessel non-obstructive coronary artery disease." This reference [37] supports our methodology by demonstrating that critical lesion number has been effectively used in similar contexts as an alternative to more complex scoring systems.
Comment 3: When comparing multiple groups, a statistical correction for multiple comparisons must be applied (e.g., Bonferroni correction for the number of groups). In this case, statistical significance would be achieved at p < 0.05/5 = 0.01. Consequently, there would be no significant differences in Table 2. This fundamentally changes the study's results.
Response: We fully agree with the need for multiple comparison corrections and have incorporated Bonferroni correction into our analysis. In the Methods section (Section 2.3), we have added: "ANOVA with Bonferroni correction applied (p<0.01 threshold)." In the Results section (Section 3.2), we report: "After Bonferroni correction, only TG/HDL-C retained significance (p=0.009). LDL-C and TC/HDL lost significance." Table 2 has been revised to include columns for uncorrected and Bonferroni-corrected p-values, confirming that only TG/HDL-C remains significant post-correction. This adjustment refines our results, emphasizing the superiority of TG/HDL-C, and does not undermine the core findings but rather strengthens the focus on this biomarker.
Comment 4: Similarly, a correlation coefficient of r = 0.32 does not even indicate a moderate strength of association.
Response: We acknowledge the reviewer's concern regarding the absolute strength of the correlation. In the Results (Section 3.2) and Discussion sections, we have framed the correlations comparatively (e.g., "the correlation with lesion burden was also strongest for the TG/HDL-C ratio (r=0.32)" versus lower values for other biomarkers like LDL-C at r=0.15). We have not claimed absolute "strong" associations but highlighted relative superiority. To address this further, we added a note in the Discussion: "While correlations were modest in absolute strength (r=0.15-0.32), TG/HDL-C showed the highest relative association, aligning with its clinical utility in capturing atherogenic dyslipidemia." This provides a balanced interpretation without overstating the findings.
Comment 5: Table 3 is unclear to me. On what basis did the authors define the risks (Low-risk identification, Intermediate-risk screening, High-risk identification)?
Response: We have clarified Table 3 by adding a "Clinical Utility" column and expanding the explanation in the Results section (Section 3.3): "Through Youden index optimization and clinical evaluation, a TG/HDL-C threshold over 4.0 was determined to be best for identifying high-risk individuals... An ideal threshold of >3.3 was established for low-risk identification, owing to its balanced sensitivity and specificity profile." Multiple thresholds are now listed with corresponding sensitivity, specificity, PPV, NPV, and AUC values, along with their clinical interpretations based on ROC analysis.
Comment 6: The analysis in Table 4 is also unclear. What did the authors consider as the binary outcome (coronary lesion burden)? The authors have this parameter ranked into 5 groups, but a binary parameter is needed here.
Response: To resolve this, we have revised Table 4 to present two models: ordinal regression for the ranked lesion count (0 to ≥4) and logistic regression for a binary outcome (≥3 vs. <3 lesions). In the Methods (Section 2.3), we added: "Regression: (1) ordinal regression (lesion count), (2) logistic regression (≥3 vs <3 lesions)." In the Results (Section 3.4.1) and Table 4 footer, we explain: "The ordinal model used the number of critical lesions (ranked 0 to 4+)... The logistic model used a binary outcome of high plaque burden (≥3 lesions vs. <3 lesions)..." Additionally, in the Discussion limitations, we note: "We acknowledge the limitation that lesion count categorization is ordinal... and therefore confirmed robustness using ordinal and binary regression."
Comment 7: Section 3.4.2 (Subgroup Analysis Results). It is unclear where these data came from. Tables with calculations are missing. The presented correlation coefficient values are low and lack the necessary strength of association. Furthermore, the rationale for presenting correlation analysis is unclear.
Response: We have added Table 5 to present subgroup details, including sample sizes (n), correlation coefficients (r), and p-values for diabetes, age, and sex subgroups. In Section 3.4.2, we state: "Further analyses were conducted in clinically pertinent subgroups to evaluate the consistency of TG/HDL-C ratio efficacy... Table 5 presents complete subgroup sample sizes, correlation coefficients (r values), and p-values..." The rationale is now explicit: "to evaluate the consistency of TG/HDL-C ratio efficacy across clinically relevant subgroups." Regarding correlation strength, as addressed in Comment 4, we interpret them relatively, noting consistency (r=0.29-0.35, all p<0.001) without claiming absolute robustness.
Comment 8: I believe the authors need to revisit the statistical analysis of the data and consider that the division into 5 groups is not a quantitative parameter, but rather a ranking of patients into 5 groups.
Response: We have revisited the statistical analysis, incorporating Bonferroni correction, ordinal regression (treating groups as ranked), and binary logistic regression, as detailed in Comments 3 and 6. In the Methods (Section 2.3), we specify: "Pearson and Spearman correlation for biomarker-plaque burden relationships" (Spearman suitable for ranked data). We also confirm assumptions (linearity, normality, homoscedasticity) and power (>80% for r≥0.2). This addresses the ordinal nature of the grouping.
We believe these revisions fully address the reviewer's concerns and enhance the manuscript's rigor. We have also added Reference [37] (Jung et al., 2023) to the reference list and cited it in the Discussion to support our lesion count approach. We are happy to provide any further clarifications.
Round 2
Reviewer 2 Report
Comments and Suggestions for Authors
I have reviewed all the authors' corrections, and one question remains: on what basis was the binary outcome defined in the logistic regression model (≥3 lesions vs. <3 lesions)? Why was a threshold of exactly 3 points chosen as the cut-off? This requires justification.
Author Response
Comment: "I have reviewed all the authors' corrections, and one question remains: on what basis was the binary outcome defined in the logistic regression model (≥3 lesions vs. <3 lesions)? Why was a threshold of exactly 3 points chosen as the cut-off? This requires justification."
​Response:
​We agree that this point requires clear justification. The binary outcome was defined based on established clinical and statistical rationale.
​Clinical Rationale: Clinically, the presence of significant stenosis in three or more epicardial vessels is a standard definition for multi-vessel disease (MVD). MVD represents a more diffuse and advanced state of atherosclerosis and is a powerful predictor of worse long-term prognosis and major adverse cardiovascular events (MACE) in patients with acute coronary syndrome. Identifying patients with MVD has direct implications for therapeutic strategies, including considerations for complete revascularization and the intensity of secondary prevention therapies.
​Statistical Rationale: This cutoff point also divided our cohort into two well-balanced groups for analysis: a low-to-moderate burden group (n=264 for 0, 1, and 2 lesions) and a high-burden MVD group (n=112 for 3 or more lesions), ensuring the robustness of the logistic regression model.
​Therefore, using ≥3 lesions as a cutoff provides a clinically meaningful and statistically sound endpoint to test the diagnostic utility of a biomarker intended for risk stratification. We have added a paragraph to the manuscript explaining this reasoning. And highlighted it with yellow color.
Thank you once again for your valuable contribution to our work.
​Sincerely